# Insomnia in Women Surviving Breast and Gynecological Cancers—A Narrative Review to Address the Hormonal Factor

**DOI:** 10.3390/cancers17244022

**Published:** 2025-12-17

**Authors:** Silvia Martella, Paola Proserpio, Maria Elena Guerrieri, Andrea Galbiati, Luigi Ferini-Strambi, Laura Cucinella, Anna Daniela Iacobone, Dorella Franchi, Rossella E. Nappi

**Affiliations:** 1Unit of Preventive Gynecology, IRCCS European Institute of Oncology, 20141 Milan, Italy; mariaelena.guerrieri@ieo.it (M.E.G.); annadaniela.iacobone@ieo.it (A.D.I.); dorella.franchi@ieo.it (D.F.); 2Department of Clinical Neurosciences, Neurology-Sleep Disorders Center, IRCCS San Raffaele Scientific Institute, 20132 Milan, Italy; paola.proserpio@outlook.com (P.P.); andrea.galbiati.unisr@gmail.com (A.G.); ferinistrambi.luigi@hsr.it (L.F.-S.); 3"Vita-Salute" San Raffaele University, 20132 Milan, Italy; 4Department of Clinical, Surgical, Diagnostic and Pediatric Sciences, University of Pavia, 27100 Pavia, Italy; 5Research Center for Reproductive Medicine, Gynecological Endocrinology and Menopause, IRCCS San Matteo Foundation, Piazzale Golgi 2, 27100 Pavia, Italy; 6Department of Public Health, Experimental and Forensic Medicine, University of Pavia, 27100 Pavia, Italy

**Keywords:** menopause, oncological menopause, premature menopause, iatrogenic menopause, cancer, breast cancer, gynecological cancer, quality of life, sleep disorders, insomnia

## Abstract

Female cancers, particularly breast and gynecological malignancies, are highly prevalent worldwide, and improved survival has shifted clinical priorities toward long-term health and survivorship care. Sleep disturbances, especially insomnia, represent a major and underrecognized burden in this population. Although insomnia affects 6–10% of the general population, its prevalence is substantially higher among cancer survivors due to cancer-related inflammation, hypothalamic–pituitary–adrenal axis hyperactivation, treatment-induced physiological changes, and psychological distress. Hormonal deprivation—often iatrogenic and occurring at a younger age—further exacerbates sleep disruption through vasomotor, cognitive, and mood alterations. Addressing insomnia is critical to mitigate downstream metabolic, cardiovascular, neurocognitive, and psychosocial consequences. Optimal management requires a multidisciplinary approach integrating oncology, gynecology, and sleep medicine. Cognitive Behavioral Therapy for Insomnia (CBT-I) remains first-line, while pharmacologic strategies, including benzodiazepines, melatonin, and menopausal hormone therapy in selected women, may be considered. Emerging neurokinin receptor antagonists show promising therapeutic potential for this population.

## 1. Introduction

Female cancers, including breast and gynecological malignancies, represent some of the most prevalent oncological conditions worldwide. Advances in screening, diagnosis, and treatment have led to significantly improved survival rates over the past few decades [1]. As a result, there is a growing population of female cancer survivors whose long-term health and quality of life are emerging as critical areas of focus in cancer care.

Despite encouraging survival outcomes, many survivors face persistent symptoms that compromise their physical, emotional, and social well-being. Among these, sleep disturbances—particularly insomnia—are strikingly common, often manifesting during active treatment and persisting long into the survivorship phase. Insomnia in cancer survivors is associated with a broad range of adverse effects, including mood disturbances, cognitive impairment, fatigue, immune dysregulation, and reduced adherence to ongoing therapies. Despite their substantial impact on patient functioning and recovery, most of these symptoms remain frequently under-recognized in routine clinical care, including in oncological practice [2].

The aim of this narrative review is to provide an integrated overview of sleep physiology, emphasizing its role in systemic homeostasis and brain–body interactions. A foundational understanding of the regulatory mechanisms governing sleep is essential to fully appreciate how cancer and its treatments disrupt these processes. The pathogenesis of insomnia in female cancer survivors will be explored from two complementary perspectives, including both neurophysiological/somnological and gynecological/oncological viewpoints. Furthermore, this review will offer practical considerations for the assessment and management of insomnia, highlighting both behavioral and pharmacologic strategies. It will also address the potential role of hormonal and non-hormonal interventions in women experiencing cancer- or treatment-induced menopause.

### Search Strategy

Although this is a narrative review and not a systematic one, we adopted a structured search strategy to ensure transparency. Relevant research was identified through PubMed/MEDLINE, Scopus, and Web of Science using combinations of terms such as insomnia, sleep disorders, sleep quality, breast cancer, gynecologic cancer, menopause, and iatrogenic menopause. The search covered primarily 2000–2024, with inclusion of earlier landmark studies when necessary. We prioritized peer-reviewed clinical studies, systematic reviews, meta-analyses, and major guidelines. Evidence was qualitatively synthesized across oncology, gynecology, endocrinology, and sleep medicine, with a focus on the mechanisms, clinical impact, and management of insomnia in female cancer survivors.

## 2. Principles of Sleep Physiology

Significant advances have been made in understanding the neural circuits that regulate sleep and wakefulness. Borbély and Achermann’s two-process model remains a key framework, proposing that sleep regulation depends on the interaction between a homeostatic process (Process S) and a circadian process (Process C) [3]. Process S reflects the accumulation of sleep pressure during wakefulness, indexed by slow-wave activity (SWA) during NREM sleep, which declines as sleep progresses. On the other hand, process C aligns the sleep–wake cycle with the external light–dark cycle, primarily through the activity of the suprachiasmatic nucleus (SCN), the central circadian clock within the brain [4,5].

Sleep initiation involves inhibition of wake-promoting systems, including serotonergic, noradrenergic, cholinergic, histaminergic, and hypocretinergic neurons. This inhibition is mediated by GABAergic neurons in the ventrolateral preoptic nucleus (VLPO), activated in part by rising extracellular adenosine during prolonged wakefulness [6]. Light entrains the SCN via melanopsin-containing retinal ganglion cells. SCN outputs coordinate circadian rhythms and influence melatonin production via the sympathetic pathway to the pineal gland, contributing to sleep timing and consolidation [7].

## 3. Functions of Sleep

Once considered a passive state, sleep is now understood as an active neurophysiological process essential for bodily maintenance, repair, and development [8]. While it contributes to neurodevelopment and memory consolidation [9,10], recent evidence highlights its critical functions in immune modulation, neuroprotection, and affective regulation [11,12].

Sleep and immune function are closely interconnected in a bidirectional manner. Indeed, during physiological slow-wave sleep, immune signaling shifts toward a pro-inflammatory state, promoting cytokine release (e.g., IL-12), T cell homing to lymphoid tissues, and the enhancement of adaptive responses. Conversely, sleep deprivation increases circulating pro-inflammatory cytokines (e.g., IL-6, TNF-α), impairs natural killer and T cell activity, and is associated with higher susceptibility to infections and chronic inflammatory diseases, like diabetes, atherosclerosis, and neurodegeneration. Immune activation, in turn, alters sleep architecture—typically increasing sleep depth and duration—as an adaptive mechanism to support host defense [11].

It is well known that sleep modulates cardiovascular function via dynamic regulation of autonomic nervous system activity, with NREM and REM stages differentially influencing heart rate variability and coronary perfusion through parasympathetic withdrawal and sympathetic activation. Sleep deprivation and disorders impair this autonomic modulation, contributing to increased risk of hypertension and cardiovascular morbidity [13].

The ability to remove metabolic waste products from the brain represents a sleep function recently discovered. Indeed, in 2012 Xie et al. firstly described the so-called “glymphatic system”, a pathway essential to maintains cerebral homeostasis by circulating cerebrospinal fluid through the brain via perivascular pathways, thus eliminating soluble proteins, metabolites, and potentially neurotoxic waste products, such as beta-amyloid, from the brain parenchyma [14]. Interestingly, glymphatic clearance is primarily active during sleep. Therefore, sleep disruption should lead to an impairment of the glymphatic system function and successively to a waste accumulation in the brain, thus favoring neurodegenerative disorders [15].

Sleep plays also a vital role in psychological well-being, emotional regulation, and stress homeostasis. Sleep disturbances are common in depression, influencing its course and prognosis [12,16,17,18,19]. In particular, insomnia is both a symptom and a risk factor for psychiatric conditions [20].

In brief, sleep disturbances may induce a condition of allostatic overload with a consequent impairment of brain plasticity, emotional, immune and endocrine pathways. Overall, this may contribute to systemic inflammation, neurodegeneration, as well as mental disorders [20].

## 4. Insomnia: Definition and Diagnosis

While sleep disorders can occur in isolation or in combination, encompassing a range of disturbances, insomnia is the most prevalent sleep disorder in the oncological population [21]. Given its high frequency, the present work primarily focuses on insomnia without ignoring the importance of assessing and treating other sleep disorders when indicated.

According to the third edition of the *International Classification of Sleep Disorders (ICSD-3)* [22], insomnia is characterized by difficulty initiating sleep, maintaining sleep continuity and early morning awakenings leading to poor sleep quality. These disturbances occur despite sufficient opportunity and appropriate conditions for sleep, leading to significant daytime impairments such as fatigue, low energy levels, reduced concentration, emotional instability, and an increased risk of accidents, particularly while driving.

Chronic insomnia is primarily a clinical diagnosis, based on patient-reported symptoms and duration criteria (with symptoms persisting ≥ 3 nights/week, ≥3 months) [22]. A detailed sleep history is the most critical component in its evaluation. Additionally, when assessing a patient with suspected insomnia, it is essential to exclude other sleep disorders, such as obstructive sleep apnea and restless legs syndrome, as this distinction is crucial for appropriate sleep-related management (Table 1).

Several validated questionnaires are available for both clinical and research purposes to identify sleep disturbances, assess their severity, and monitor treatment response (Table 2).

A daily sleep diary is also frequently recommended and is considered the gold-standard subjective measure for the diagnosis of chronic insomnia, as it provides a more accurate and real-time record of a patient’s sleep–wake patterns [23].

In addition to subjective measures, sleep can be assessed using objective tools such as actigraphy and polysomnography. Actigraphy involves the use of a wrist-worn device that tracks motor activity, enabling the estimation of various sleep parameters, including total sleep time, sleep onset latency, wake after sleep onset, and sleep efficiency [23]. Polysomnography (PSG) remains the gold standard for objectively assessing sleep architecture in a laboratory setting. While PSG is not necessary for diagnosing insomnia itself, it is indicated when other sleep disorders, such as periodic limb movement disorder (PLMD), obstructive sleep apnea, or parasomnia, are suspected. Additionally, PSG should be considered in patients with insomnia that is refractory to treatment, those at risk for fatigue-related accidents, and anytime there is a significant discrepancy between a patient’s subjective experience of sleep and objective sleep measurements [23].

## 5. Prevalence of Insomnia in Female Cancer Survivors

Insomnia represents the most prevalent sleep disorder in the general population, with an estimated prevalence ranging between 6% and 10%, and exhibits a higher incidence in women compared to men [24]. This gender disparity may partially account for the higher incidence of insomnia observed among patients with female cancers.

A recent systematic review and meta-analysis analyzed 59 studies encompassing a total of 16,223 cancer patients and reported that 57.4% of individuals experienced poor sleep quality. Among the included studies, three case–control investigations demonstrated a threefold higher prevalence of poor sleep quality in cancer patients compared to healthy controls. Notably, sleep disturbances appeared to vary by cancer type, with patients affected by gynecological cancers displaying some of the highest prevalence rates [25].

Specifically, insomnia affects 14–60% of patients with ovarian cancer and is associated with increased morbidity, emotional distress, anxiety, depression, and reduced quality of life [26,27]. Sleep disturbances often persist throughout the disease trajectory and are not adequately addressed by pharmacological treatments alone. Emerging evidence also indicates that insomnia may be linked to a higher risk of invasive serous ovarian cancer, whereas good sleep quality could be protective for this subtype [28].

Regarding endometrial cancer, large pooled analyses do not support a strong association between night-shift work, sleep duration, or sleep-related traits and overall endometrial cancer risk in postmenopausal women [29,30]. Nonetheless, sleep disorders are common among patients with gynecologic malignancies and may exacerbate symptom burden while diminishing quality of life. Data on cervical cancer, however, remain very limited.

The most recent meta-analysis focusing specifically on breast cancer survivors reported an overall prevalence of poor sleep quality of 62%. Within this cohort, insomnia severity was categorized as subthreshold (29%), moderate (24%), severe (4%), and non-specified (33%) [31]. Subgroup analyses indicated a significantly higher prevalence of sleep disturbances during active treatment phases compared to pre-treatment or post-treatment periods [31]. A recent longitudinal investigation in a heterogeneous cancer population revealed that insomnia rates declined only marginally after one year [32]. This occurs despite 80% of patients being tumor-free and exhibiting improvements in psychological parameters [32]. Thus, sleep disturbances in oncologic populations may persist independently of tumor status or emotional recovery.

Nonetheless, it remains methodologically challenging to accurately estimate the true prevalence of sleep disturbances in patients with female cancers. Considerable heterogeneity exists across epidemiological studies due to variations in sampling strategies and assessment methods. In particular, the timing of evaluation represents a significant source of inconsistency, as sleep disturbances may be assessed at diagnosis, during oncologic treatment (e.g., chemotherapy, radiotherapy, or endocrine therapy), or during survivorship. Moreover, most studies rely on subjective assessment tools—primarily self-reported questionnaires or structured interviews—while objective sleep measurements such as PSG or actigraphy are infrequently employed. Many studies also utilize either cross-sectional designs or longitudinal designs initiated after cancer diagnosis, thereby precluding any determination of pre-existing sleep disturbances [33].

Finally, risk factors associated with insomnia in cancer patients are not yet fully defined. Among breast cancer survivors, hot flashes, non-Caucasian ethnicity, and postmenopausal status were identified as significant correlates of increased insomnia risk in a meta-analysis [33]. Further subgroup analyses in studies with high heterogeneity revealed that pain, depressive symptoms, and fatigue were also independently associated with poor sleep outcomes [33].

## 6. Pathophysiology of Sleep Disorders in Female Cancer Survivors

Insomnia in female cancer patients arises from a multifactorial pathogenesis involving cancer-related aspects, treatment-related effects, psychological distress (e.g., anxiety, depression), dysfunctional cognitive processes (e.g., sleep-related worry) and behavioral factors (maladaptive sleep habits) [34]. In addition, hormonal disruptions, especially those related to treatment-induced menopause, play a central role in promoting and sustaining sleep disturbances [35]. Furthermore, emerging evidence suggests a role for genetic and neurobiological factors contributing to pre-diagnosis vulnerability [36].

### 6.1. Inflammatory and Neuroendocrine Mechanisms

Cancer typically induces systemic inflammation, which, as previously discussed, can contribute to sleep disturbances through multiple interrelated mechanisms. Breast cancer and its treatments (chemotherapy, radiotherapy, endocrine therapy) are associated with both systemic and localized inflammatory responses, marked by elevated levels of pro-inflammatory cytokines such as interleukin-6 (IL-6), tumor necrosis factor-alpha (TNF-α), and C-reactive protein (CRP) [37]. These cytokines have been implicated in increased sleep latency, diminished slow-wave and REM sleep, and disrupted non-REM (NREM) sleep architecture [38,39].

Chronic low-grade inflammation, often sustained by sleep deprivation, is closely linked to dysregulation of the HPA axis, further amplifying neuroendocrine disruption. Cortisol plays a key role in the pathophysiology of insomnia promoting maladaptive stress responses in breast cancer patients. Indeed, both acute and chronic psychological stressors—alongside cancer-related factors -can result in altered cortisol secretion patterns, including elevated evening and nocturnal cortisol levels, a flattened diurnal slope, and abnormal nocturnal surges. Some studies [40,41] have demonstrated that prolonged cortisol elevation is associated with delayed sleep onset, increased nighttime awakenings, and reduced overall sleep efficiency.

It is worth mentioning that sex-related differences modulate the impact of cancer-associated inflammation on sleep. Females exhibit more pronounced inflammatory responses—likely influenced by their pattern of sex hormones—which may contribute to the higher prevalence of insomnia and poorer sleep quality in cancer population [42].

Finally, the immune response in cancer patients also contributes to a broad range of physical and psychological symptoms, including fatigue and depression, both of which are highly prevalent [43]. These symptoms further activate the HPA axis and exacerbate cortisol dysregulation, perpetuating a vicious cycle that sustains and worsens insomnia in oncologic populations [43].

### 6.2. Treatments-Related Effects

Among different cancer treatments (chemotherapy, radiotherapy, surgery, and endocrine therapy), chemotherapy is associated with a higher prevalence and severity of insomnia symptoms during active treatment. Studies consistently show that patients undergoing chemotherapy report significantly worse sleep quality and more severe insomnia compared to those not receiving chemotherapy or those in other treatment phases [44,45].

The underlying mechanisms are multifactorial. Chemotherapy has been shown to alter circadian regulation and immune system functioning, contributing to dysregulation of sleep–wake cycles. In addition, it commonly induces a range of somatic symptoms—including nausea, pain, fatigue, and vasomotor disturbances such as hot flashes—that serve as mediators of sleep disruption. Concurrently, the treatment phase is often characterized by increased psychological distress, including heightened levels of anxiety and depressive symptoms, both of which are strongly associated with insomnia in oncologic populations [46].

### 6.3. Psychological and Behavioral Mechanisms

Cancer-related stress—such as fear of recurrence—has been frequently reported by patients as a primary contributor to sleep difficulties. One study found that 87% of patients attributed their insomnia to stress following cancer diagnosis [47]. Accordingly, interventions that directly target dysfunctional cognitions have received particular attention in the literature, given their pivotal role in the maintenance of chronic insomnia [48].

However, numerous studies have identified mood disorders as independent predictors of both the onset and persistence of sleep disturbances. Insomnia frequently co-occurs with depression and anxiety, independently from other cancer-related or physical symptoms [49]. Elevated mood symptoms are associated with greater insomnia severity, irrespective of cancer stage or treatment modality. Longitudinal data indicate a bidirectional relationship, where worsening depressive symptoms predict deteriorating sleep quality and vice versa [50,51]. These symptoms often persist beyond the acute treatment phase, suggesting that mood disturbances may perpetuate insomnia even after resolution of the initial oncological stressors [32]. The underlying mechanisms are multifactorial and include psychological distress, maladaptive coping, and neurobiological alterations related to both cancer and its treatment. Similar findings in women with metastatic breast cancer further support the predictive role of baseline depression and stress in the development of sleep disturbances [52].

## 7. The Importance of Menopause and Neuroendocrine System

Cancer therapies are frequently associated with the onset of early or premature menopause [53,54,55,56]. Several chemotherapeutic agents exhibit high gonadotoxic potential and can induce irreversible ovarian failure, especially in women over the age of 40 or those with reduced ovarian reserve. Similarly, pelvic and spinal radiotherapy may result in permanent ovarian damage [55,56]. Surgical interventions, such as bilateral oophorectomy, are commonly performed as part of treatment in gynecological cancers and risk-reducing strategies in breast cancer women carrying BRCA mutations. This procedure leads to immediate and irreversible menopause due to the abrupt cessation of ovarian steroidogenesis [57]. Finally, endocrine therapies for hormone receptor–positive tumors, particularly breast cancer, often induce prolonged ovarian suppression. In premenopausal women, GnRH analogs combined with tamoxifen or aromatase inhibitors can lead to sustained hypoestrogenism, sometimes persisting until natural menopause [58,59].

Sleep disorders are frequently reported in women during natural menopause even if it is unclear whether menopause physiologically contributes to circadian rhythm aging that may derive from reduced endogenous melatonin secretion [60]. Interestingly, Song et al. studied a sample of 464 breast cancer patients compared to a control group and identified alterations in four SNPs in genes responsible for circadian rhythm control, suggesting a reciprocal and not unilateral relationship between sleep disturbances and oncogenesis, even in the premenopausal period [61].

However, the intensity and frequency of sleep disorders seem to be higher in patients with iatrogenic menopause. Indeed, these women experience more severe and frequent vasomotor symptoms (VMSs) (hot flashes, night sweats), and nocturnal episodes are strong predictors of sleep disruption [62].

In a group of BRCA mutation carriers who underwent prophylactic oophorectomy, the prevalence of VMSs rose from 6% at baseline to 59% at 24 months, while the prevalence of night sweats increased from 21% to 39% [63].

A study including 934 women with a history of cancer, primarily breast and gynecological cancers, reported a mean of 6 hot flashes per day in cancer survivors, compared to 3 in the healthy menopausal controls; survivors were more likely to experience more than 10 episodes per day [64].

Cognitive symptoms, mood disturbances, sexual dysfunction and genitourinary symptoms (vaginal dryness, dyspareunia, urinary symptoms) are also more common and severe after abrupt ovarian failure compared to natural menopause and could further increase the risk of insomnia [65,66].

In breast cancer patients, VMSs are a key mediator between iatrogenic menopause and insomnia. Among breast cancer survivors undergoing adjuvant endocrine therapy, 70–80% reported VMSs, with a frequency three times higher than in healthy postmenopausal controls, and some of them experienced episodes lasting more than 10 min [67,68]. Multiple cross-sectional and longitudinal studies have identified VMS severity and frequency as independent predictors of insomnia, with odds ratios for sleep disturbance ranging from 2.0 to 2.3 [69]. Longitudinal data also show that increases in VMSs correlate temporally with worsening sleep, and that VMS management improves sleep outcomes [44,70]. Savard et al. demonstrated in a sample of breast cancer patients that quality of hot flashes (e.g., onset speed and duration) has a greater impact on insomnia than frequency alone. Objective measures (sternal skin conductance, PSG) confirm that hot flashes, particularly during their onset and preparatory phases, disrupt sleep architecture [71]. Notably, breast cancer survivors show altered circadian patterns of hot flashes compared to healthy postmenopausal women [60].

The mechanism likely involves night sweats, which cause repeated awakenings and impair sleep continuity attributable to altered central thermoregulation processes related to estrogen deprivation. This new temperature set-point results from increased norepinephrine secretion in the central nervous system (CNS) and sympathetic activation in the anterior cingulate cortex [71].

However, some patients with breast cancer report insomnia also in the absence of hot flashes. Indeed, meta-analyses and systematic reviews indicate that menopause represents a significant risk factor for insomnia, also after adjusting for VMSs, indicating a direct role of the neuroendocrine changes associated with menopause on sleep architecture. While there has been little research investigating the direct impact of endocrine therapy on sleep, levels of both estrogens and progesterone have been shown to independently impact sleep–wake regulation via CNS pathways [72]. Estrogens modulate serotonergic and GABAergic neurotransmission, which are critical for sleep initiation and maintenance, and also influence circadian rhythm regulation [73,74]. Estrogen deficiency causes KNDy neuron hyperactivity, increasing SP and NKB expression, which disrupts temperature regulation and influences neurotransmitter release (namely serotonin, dopamine), further contributing to sleep disturbances. On the other hand, progesterone has sedative properties mediated by its metabolites, acting as positive allosteric modulators of GABA A receptors and thus promoting sleep continuity and reducing sleep latency [75].

Weight gain-related factors, as well as low physical activity associated with fatigue or arthralgia, may also contribute to sleep disturbances in menopausal women with cancer [76]. For instance, in a study investigating 100 first stage endometrial cancer survivors, those who reported severe obesity were more susceptible to poor sleep and depression [77].

Disfigurement and altered body image affect mood, sleep, and overall quality of life in young female cancer survivors [78]. Sleep disturbances could also be favored by sexual complaints associated with the genitourinary syndrome of menopause, especially in breast cancer survivors under endocrine adjuvant treatment or gynecological patients after radiation [79,80,81]. The role of antidepressants, often prescribed to cancer survivors with mood disorders, warrants attention due to their potential impact on sexual function [55].

The summary of the causes of insomnia, including menopause-related factors, are reported in Figure 1.

The figure illustrates the multifactorial pathways linking cancer and menopause-related factors to the development of insomnia in women with cancer. Genetic and neurobiological factors, cancer-related variables (including treatments, body image concerns, and systemic chronic inflammation), psychological and behavioral factors, and immunological responses interact bidirectionally, creating a complex network that predisposes to and maintains insomnia. Hormonal alterations, particularly activation of the hypothalamic–pituitary–adrenal axis by stressors and the sex-specific impact of inflammatory and immune mediators, are central mechanisms, with menopause acting as a key modulator. Menopause, especially when early, premature, or iatrogenic and accompanied by sex hormone deprivation, contributes to insomnia through several downstream consequences: neuronal modulation (involving KNDy neurons and neurotransmitters such as serotonin, dopamine, and GABA), vasomotor symptoms (hot flushes and night sweats) with associated changes in mood, cognition, and “brain fog”, and other menopausal issues (including dysmetabolism, weight gain, genitourinary syndrome, sexual dysfunction, fatigue, and additional symptoms). Altogether, these interconnected biological, psychological, and treatment-related factors reinforce each other to perpetuate insomnia in women with cancer across the menopausal transition.

## 8. Impact of Sleep Disorders in Female Cancer Survivors

Beyond the immediate effects of sleep disturbances on mood, fatigue and daytime functioning, these disorders exert a profound influence on emotional, cognitive, and physical health domains, contributing to a cumulative burden that may compromise long-term recovery and quality of life [82,83].

Emotional well-being is particularly vulnerable to the effects of chronic insomnia. Mood disturbances often persist long after the conclusion of active treatment, suggesting that sleep alterations are not merely reactive to acute stress, but may represent a chronic consequence of the oncological journey [84].

Sleep disturbances also intersect with sexual health, body image, and intimacy and these effects are especially prominent in younger survivors who may already face significant psychosocial challenges in the aftermath of treatment and menopause [85,86].

Cognitive complaints are also frequently reported by female cancer survivors with insomnia, including difficulties with attention, memory, and executive function [87].

A study of 1072 breast cancer survivors revealed that 79% reported cognitive symptoms, which were particularly prevalent among patients with insomnia, as well as in those younger than 55 years, with a history of chemotherapy, or comorbid mood disorders [88]. While these symptoms are often perceived as subjective or multifactorial, growing evidence points to objective alterations in sleep-dependent neurophysiological processes. Changes in sleep architecture—reduced slow-wave sleep, increased nocturnal arousals, and heightened cortical activity—have been linked to measurable declines in attention, information processing speed, and emotional regulation [89]. This suggests that insomnia may contribute to real cognitive deterioration, especially in vulnerable populations such as postmenopausal or medically induced menopausal women.

From a systemic perspective, disrupted sleep has been associated with autonomic imbalance, metabolic alterations, and increased inflammatory activity. Insomnia may contribute to weight gain, insulin resistance, and dyslipidemia, thereby exacerbating cardiovascular and metabolic risk profiles already heightened by cancer treatments and menopausal status [90,91].

Finally, sleep disturbances have practical and economic implications. Gonzalez et al. showed that sleep problems significantly mediate the relationship between cancer diagnosis and increased healthcare costs and work absenteeism [92]. Poor sleep quality is also the leading cause of early discontinuation of adjuvant aromatase inhibitor therapy, further highlighting the critical need to address sleep in cancer care [93] (Table 3).

## 9. Assessment of Sleep Disorders in Female Cancer Survivors

Given the high prevalence of sleep disturbances in women with gynecological or breast cancer, proper assessment tools are essential. Recognizing this need, major oncology organizations such as the National Cancer Institute (NCI), National Comprehensive Cancer Network (NCCN), and European Society for Medical Oncology (ESMO) recommend routine screening for insomnia at multiple stages of cancer care, including diagnosis, treatment, and survivorship [2].

Despite these recommendations, insomnia remains underassessed in clinical settings. Studies show that even when patients report sleep problems, these are often undocumented or insufficiently addressed by clinicians [94,95,96]. A U.S. survey of cancer centers found that most institutions screened less than 25% of survivors for sleep disturbances, and only few offered evidence-based treatments. A lack of training was cited as a major barrier among clinicians [95].

To address this, the NCCN proposes a simple yet effective screening question: *“Are you having trouble falling asleep, staying asleep, or waking up too early?”* [97]. Though recommended, this question is rarely integrated into routine care. Feasibility studies suggest that brief self-report questionnaires administered during follow-up visits can efficiently identify overlooked sleep issues [98].

Additionally, insomnia is included in the Edmonton Symptom Assessment System (ESAS), a validated, multidimensional tool widely used in oncology. Its brevity and availability in multiple languages make it suitable for routine screening across diverse patient populations [99].

## 10. Treatment of Insomnia in Female Cancer Survivors

Treatment of insomnia in women with gynecologic or breast cancer includes both non-pharmacologic and pharmacologic options. The therapeutic choice depends on clinical features, patient preference, access to behavioral therapies, and clinician expertise (Figure 2). It is crucial to first address contributing factors such as hot flashes, pain, or nocturia.

Cognitive Behavioral Therapy for Insomnia (CBT-I) is recommended as a first-line treatment for chronic insomnia in all adults, including those with comorbid medical conditions [23]. It integrates components like psychoeducation, sleep hygiene, sleep restriction, stimulus control, and cognitive restructuring, usually across 4–8 sessions with trained professionals. Delivery formats include in-person, telephone, and digital platforms, with electronic CBT-I showing comparable efficacy to face-to-face treatment [100].

Two meta-analyses confirmed the efficacy of CBT-I in cancer patients. Gao et al. found short-term benefits in sleep onset latency, wake after sleep onset, and sleep efficiency, though long-term effects were less sustained [101]. Squires et al. reported more persistent improvements in insomnia severity, as well as reductions in fatigue, depression, and anxiety [102]. Specifically for breast cancer survivors, a meta-analysis of 14 randomized controlled trials involving a total of 1363 women found that CBT-I significantly reduces insomnia and improves sleep quality, with medium-to-large effects immediately post-intervention, sustained benefits in the short term, and effects still evident up to 12 months [103]. Similarly, a randomized controlled trial of 35 women undergoing surgery for gynecological cancer reported that, compared with psychoeducation, CBT-I was associated with increased sleep efficiency, as well as reduced total wake time and sleep onset latency [104].

Mindfulness-based therapies (MBTs) have also shown promise, particularly in breast cancer patients, improving sleep quality with variable effect sizes [105,106]. Other non-pharmacologic approaches—physical activity, yoga, acupuncture—have shown preliminary benefits, though current evidence remains limited [107,108,109,110].

Despite guideline preference for behavioral treatments, hypnotics remain commonly used. One retrospective study found that 26% of breast cancer patients received hypnotics in the first post-diagnosis year; 17% were new users, and 4% progressed to chronic use [111]. Another study during chemotherapy showed that 32.3% were prescribed sleep medications, mostly lorazepam and zolpidem [112].

Pharmacologic treatment is less well defined in cancer populations, with few specific guidelines. Current general recommendations endorse short-term use of benzodiazepines or Z-drugs for acute insomnia or as second-line for chronic insomnia after CBT-I [23]. In patients with insomnia and hot flashes, SSRIs/SNRIs, gabapentin, and clonidine have shown benefit [113,114,115].

Although the evidence on efficacy remains uncertain and objective sleep measures are less consistently improved, melatonin (2 mg prolonged release) is recommended for adults over 55. The most recent Cochrane review and the New England Journal of Medicine clinical update [116,117] highlight that the certainty of evidence supporting melatonin’s benefits on sleep in cancer patients remains very low. Indeed, they found that randomized controlled trials are limited by small sample sizes, high risk of bias, and indirectness, with only a few studies specifically addressing breast cancer populations. Improvements in subjective sleep quality have been reported in some trials, but objective sleep measures and quality of life outcomes remain inconsistent and uncertain. However, emerging evidence suggests that melatonin may also reduce chemotherapy-related toxicity and possess anticancer properties [118,119].

Dual orexin receptor antagonists (DORAs)—including suvorexant, lemborexant, and daridorexant—have been recently approved by FDA for the treatment of adult insomnia. They act by selectively antagonizing orexin receptors (OX1R and OX2R), thereby reducing wakefulness and facilitating sleep [120]. Clinical trials and meta-analyses demonstrate that DORAs significantly improve sleep latency and maintenance, with a favorable safety profile and minimal next-day residual effects. However, although the literature highlights theoretical interest in targeting orexin receptors for cancer therapy [121], to date no clinical studies have been conducted on dual orexin receptor antagonists (DORAs) specifically in patients with female cancers.

Other perspectives should be considered, including menopause hormone therapy (MHT) and the novel NKR antagonists. Since menopause is associated with sleep disorders, MHT may also be considered for cancer survivors experiencing climacteric symptoms (in particular, VSMs) or who have undergone premature menopause [122]. However, in oncological patients more than in healthy individuals, risk assessment must consider tumor histotype, disease stage, ongoing treatments, and associated comorbidities. While MHT is contraindicated for breast cancer survivors, it is generally not contraindicated per se for most gynecological cancers, although the therapy must always have a clear indication [123,124,125,126,127,128].

It remains challenging to establish a clear distinction between the different molecules used as MHT and their impact on sleep, particularly in oncological patients. However, a systematic review has indicated that the formulations and routes of administration of MHT may influence the effect size. Indeed, evidence supports the use of transdermal estradiol combined with micronized progesterone for at least 6 months in menopausal women experiencing sleep disorders [129]. Micronized progesterone, especially when administered orally, has a well-established effect on the sleep cycle, because it acts as a neurosteroid. It is also important to underline that micronized progesterone has a strong safety profile which is highly relevant in an oncological population [130].

Considering the potential link between sleep disorders and sexual dysfunction, managing genitourinary syndrome—especially in women receiving adjuvant endocrine therapy for breast cancer—may improve sexual function, mood, and sleep, thereby generating a positive domino effect.

Moreover, in women with hormone-responsive cancer, or those who have contraindications to MHT or do not wish to use it, some herbal treatments (pollen extract, black cohosh) or SSRI/SNRI drugs may alleviate sleep disturbances and vasomotor symptoms, without any hormonal effects [131,132]. For patients without contraindications to phytoestrogens, therapy with soy isoflavones could alleviate hot flash symptoms and thus improve sleep [133,134].

Promising data have emerged regarding the use of NKR antagonists, which have shown a significant impact on reducing VMS, improving sleep, and enhancing overall quality of life [135,136,137]. Their use in women with hormone responsive cancer remains off-label. However, in women with previous breast cancer under endocrine treatment, a recently published phase III randomized trial with the NK1-3R antagonist (elinzanetant) confirmed its efficacy reducing the frequency of hot flashes from 11.5 to 6.5 per week at 4 weeks, with a maintained safety profile up to 12 weeks and in the extended 52-week regimen [138]. Elinzanetant has been recently approved by FDA and EMA. Another drug of the same class, the NK3-R antagonist fezolinenant, has been tested in women unsuitable for MHT showing efficacy and safety [139], but data published specifically in women with breast cancer are not yet available.

Drawing on the evidence summarized above and in line with current guideline recommendations, oncologists should first be familiar with the most prevalent sleep disorders in patients with cancer, particularly insomnia. This knowledge is essential because, in routine clinical practice, oncologists should systematically assess the presence of sleep disturbances in their patients. When adequately trained and confident in managing these conditions, oncologists may initiate first-line interventions for uncomplicated insomnia, while referring patients with complex or treatment-refractory insomnia, as well as those with suspected alternative sleep disorders, to a sleep specialist. Ideally, a sleep expert should be embedded within the multidisciplinary oncology team; when this is not feasible, there should at least be a designated sleep center to which patients can be referred for comprehensive assessment and management.

## 11. Conclusions

Sleep disturbances—particularly insomnia—are highly prevalent among female cancer survivors and exert a significant impact on psychological, cognitive, endocrine, and immune function, yet they are often underrecognized and undertreated by both clinicians and patients. While advancements in cancer therapies have markedly improved survival, the quality of that survival remains a major challenge, often compromised by persistent, underestimated symptoms such as poor sleep.

Understanding the physiological basis and systemic role of sleep is essential for interpreting the pathogenesis of insomnia in oncologic contexts. As discussed in this review, sleep regulation is deeply interconnected with neuroendocrine, immunological, and psychological systems—all of which are commonly affected by cancer and its treatments. A comprehensive approach to insomnia in female cancer survivors must therefore be multidisciplinary, involving collaboration among oncologists, gynecologists, neurologists, psychotherapists or sleep specialists. The latter can provide critical expertise in the interpretation of sleep-related complaints and in tailoring individualized treatment strategies.

It is important to acknowledge the limitations of the current evidence in women with gynecological cancers because data are predominantly available in breast cancer survivors.

The analysis of sleep disturbances in gynecologic cancer survivors is hindered by a lack of longitudinal and interventional studies, a limited understanding of the biological mechanisms linking sleep disturbances to cancer progression, and the absence of effective therapeutic interventions.

Sleep-related outcomes in gynecologic malignancies—such as ovarian, endometrial, and cervical cancers—should be better explored because existing studies mainly rely on subjective reporting and heterogeneous methodologies.

To conclude, there is a significant need for further data in this area of research using standardized assessment tools in longitudinal study designs. Such investigations seem essential not only to improve our understanding of the impact of insomnia on the health and quality of life of cancer survivors but also to identify potential pre-existing vulnerabilities. In addition, identification of common clues among individuals with sleep disturbances and cancer could help clarify the complex interplay between hormonal states, cancer, and sleep alterations.

## Figures and Tables

**Figure 1 cancers-17-04022-f001:**
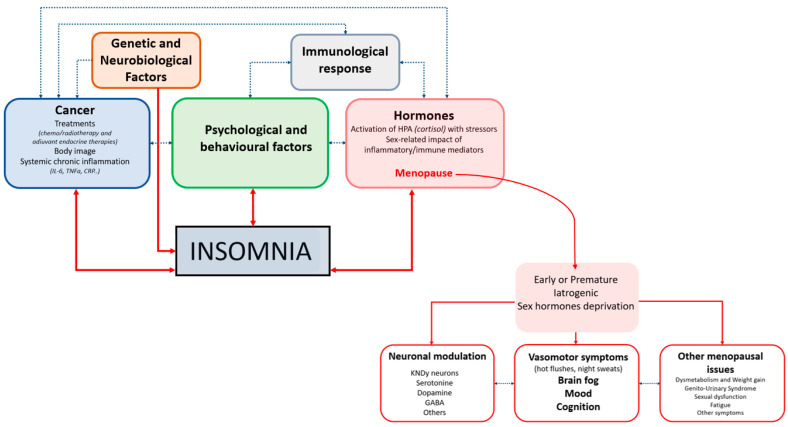
Pathophysiology Of Sleep Disorders In Female Cancer Survivors.

**Figure 2 cancers-17-04022-f002:**
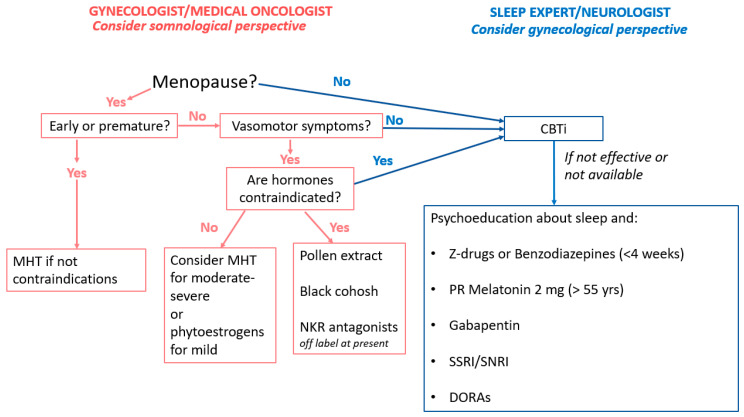
Management Of Sleep Disorders In Female Cancer Survivors.

**Table 1 cancers-17-04022-t001:** Other common sleep disorders.

Sleep Disorder	Sign/Symptom	Testing	Treatment
Obstructive Sleep Apnea	Snoring, witnessed apneas, gasping or choking, nocturia, excessive daytime sleepiness	Polysomnography/ Cardiorespiratory monitoring	Continuous positive airway pressure Oral appliance Surgery
Restless Leg Syndrome	Uncomfortable sensation in both legs, symptoms are worse in the evening, improve with movement	History Iron/ferritin	Dopaminergic agonists Gabapentin/Pregabalin Clonazepam Opioid
Circadian Rhythm Disorders	Early or late onset and wake-up time	History Sleep diary Actigraphy	Melatonin Bright light therapy
Parasomnia	Complex motor behavior during sleep	History Video Polysomnography	Clonazepam Melatonin
Narcolepsy	Excessive daytime sleepiness, cataplexy, sleep paralysis and hallucinations	Polysomnography Vigilance tests	Modafinil Sodium oxybate Pitolisant Solriamfetol

**Table 2 cancers-17-04022-t002:** Principal validated questionnaires assessing insomnia.

Questionnaires	
**Insomnia Severity Index (ISI)**	Self-report seven-item scale investigating sleep over the past 14 days (difficulties initiating sleep, staying asleep, and early morning awakenings, satisfaction with current sleep pattern, interference with daily functioning, noticeability of impairment attributed to the sleep problem, and degree of distress or concern). Each item is scored on a four-point scale (score range 0–28). A score ≥8 indicates subthreshold insomnia and a score ≥15 indicates clinical insomnia.
**Sleep Condition Indicator (SCI)**	Eight-item scale. The items assess sleep disturbances and sleep-related daytime functioning over the previous month. Each item is scored on a five-point scale (0–4). Possible total score ranges from 0 to 32, with higher values indicative of better sleep. A score ≤ 16 is considered a ‘probable insomnia disorder’.
**Pittsburgh Sleep Quality Index (PSQI)**	Self-administered questionnaire that measures sleep disturbance and usual sleep habits during the prior month. Nineteen individual items generate seven “component” scores: subjective sleep quality, sleep latency, sleep duration, habitual sleep efficiency, sleep disturbances, use of sleeping medication, and daytime dysfunction. A higher global score (sum of the seven domains, score range 0–21) indicates worse sleep quality. A score of >5 indicates poor sleepers.

**Table 3 cancers-17-04022-t003:** Impact of sleep disorders in female cancers survivors.

Mood Disorders (Anxiety, Depression)
Cognitive impairment and neurodegenerative diseases
Sexual dysfunction
Physical impairment and higher metabolic and cardiovascular risk
Worsening of social function
Healthcare cost and work absenteeism
Uncertain role in carcinogenesis

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
