# Peer review of "Insomnia in Women Surviving Breast and Gynecological Cancers—A Narrative Review to Address the Hormonal Factor"

_cancers, 2025, doi:10.3390/cancers17244022_

Round 1
Reviewer 1 Report
Comments and Suggestions for Authors
I was please to review the manuscript entitled “Insomnia in women surviving breast and gynecological cancers – a narrative review to address the hormonal factor”. The topic is clinically relevant and aligns with the increasing focus on survivorship and quality of life in oncology. The review is thorough, well referenced, and written with strong academic rigor. It successfully integrates perspectives from sleep medicine, gynecology, oncology, and neuroendocrinology. Below, I offer suggestions aimed at improving clarity, structure, and methodological transparency, while acknowledging the overall high quality of the work.
- The English is generally strong, but a few sections contain very long sentences that could be simplified for clarity. A light copy-edit would help reduce sentence complexity and improve readability.
- As a narrative review, methods are not expected to follow a systematic protocol; however, clarity could be improved by adding: A brief description of search strategy (databases, keywords, time frame) and a note about how evidence was prioritized or synthesized. Adding this section would enhance transparency and rigor.
- Section 6 provides an excellent overview but is somewhat dense.
A clearer sub-structuring (e.g., “Inflammatory mechanisms,” “Neuroendocrine mechanisms,” “Psychological and behavioral mechanisms”) would greatly enhance readability and highlight the multidimensional nature of insomnia. - The authors rightfully acknowledge that most available data are derived from breast cancer populations. However, since the manuscript title explicitly refers to "gynecological cancers," I encourage the authors to expand the dedicated content on ovarian, endometrial, and cervical cancers and more clearly state the knowledge gaps and research priorities in these populations
- On Figure 1 a more detailed explanation of the components illustrated and their interactions would help readers grasp the key causal pathways.
- Please, ensure that all abbreviations used in the text appear in the abbreviations list.
- The section on treatment is informative and balanced. However, the review would benefit from a more explicit clinical perspective. When should oncologists screen or refer patients to sleep specialists? How should sleep disturbances be integrated into survivorship care pathways?
- The authors discuss melatonin, DORAs, MHT, phytoestrogens, and NK receptor antagonists. However, It would be helpful to clearly distinguish which therapies have evidence specifically in cancer survivors and where data are extrapolated from non-oncologic populations
Author Response
I was pleased to review the manuscript entitled “Insomnia in women surviving breast and gynecological cancers – a narrative review to address the hormonal factor”. The topic is clinically relevant and aligns with the increasing focus on survivorship and quality of life in oncology. The review is thorough, well referenced, and written with strong academic rigor. It successfully integrates perspectives from sleep medicine, gynecology, oncology, and neuroendocrinology. Below, I offer suggestions aimed at improving clarity, structure, and methodological transparency, while acknowledging the overall high quality of the work.
Comment 1:
- The English is generally strong, but a few sections contain very long sentences that could be simplified for clarity. A light copy-edit would help reduce sentence complexity and improve readability.
We tried to improve our English for the sake of clarity.
Comment 2:
- As a narrative review, methods are not expected to follow a systematic protocol; however, clarity could be improved by adding: A brief description of search strategy (databases, keywords, time frame) and a note about how evidence was prioritized or synthesized. Adding this section would enhance transparency and rigor.
Thank you for your suggestion. We added this paragraph to improve methodological transparency and strengthen the rigor of the narrative review.
Search Strategy
Although this is a narrative review and not a systematic one, we adopted a structured search strategy to ensure transparency. Relevant literature was identified through PubMed/MEDLINE, Scopus, and Web of Science using combinations of terms such as insomnia, sleep disorders, sleep quality, breast cancer, gynecologic cancer, menopause, and iatrogenic menopause. The search covered primarily 2000–2024, with inclusion of earlier landmark studies when necessary. We prioritized peer-reviewed clinical studies, systematic reviews, meta-analyses, and major guidelines. Evidence was qualitatively synthesized across oncology, gynecology, endocrinology, and sleep medicine, with a focus on mechanisms, clinical impact, and management of insomnia in female cancer survivors.
Comment 3:
- Section 6 provides an excellent overview but is somewhat dense.
A clearer sub-structuring (e.g., “Inflammatory mechanisms,” “Neuroendocrine mechanisms,” “Psychological and behavioral mechanisms”) would greatly enhance readability and highlight the multidimensional nature of insomnia.
Thank you for your suggestion. We added these sub-sections: Inflammatory and neuroendocrine mechanisms, Treatments-related effects and Psychological and behavioral mechanisms.
Comment 4:
- The authors rightfully acknowledge that most available data are derived from breast cancer populations. However, since the manuscript title explicitly refers to "gynecological cancers," I encourage the authors to expand the dedicated content on ovarian, endometrial, and cervical cancers and more clearly state the knowledge gaps and research priorities in these populations
Thank you for this additional suggestion. We have incorporated further information on ovarian, endometrial, and cervical cancers to provide more precise description and clear reference to gynecologic neoplasms. Despite data on gynecological cancers and insomnia remain very limited, we truly appreciate the suggestion to emphasize the need for future research in this area.
In paragraph “PREVALENCE OF INSOMNIA IN FEMALE CANCER SURVIVORS”
[.. Specifically, insomnia affects 14–60% of patients with ovarian cancer and is associated with increased morbidity, emotional distress, anxiety, depression, and reduced quality of life. Sleep disturbances often persist throughout the disease trajectory and are not adequately addressed by pharmacological treatments alone. Emerging evidence also indicates that insomnia may be linked to a higher risk of invasive serous ovarian cancer, whereas good sleep quality could be protective for this subtype. Regarding endometrial cancer, large pooled analyses do not support a strong association between night-shift work, sleep duration, or sleep-related traits and overall endometrial cancer risk in postmenopausal women. Nonetheless, sleep disorders are common among patients with gynecologic malignancies and may exacerbate symptom burden while diminishing quality of life. Data on cervical cancer, however, remain very limited.]
In paragraph “CONCLUSIONS”
[The analysis of sleep disturbances in gynecologic cancer survivors is hindered by a lack of longitudinal and interventional studies, a limited understanding of the biological mechanisms linking sleep disturbances to cancer progression, and the absence of effective therapeutic interventions.]
The bibliographic references have been appropriately updated.
Comment 5:
- On Figure 1 a more detailed explanation of the components illustrated and their interactions would help readers grasp the key causal pathways.
We followed your suggestion and modified caption of figure 1.
The figure illustrates the multifactorial pathways favoring the development of insomnia in women with cancer. Genetic and neurobiological factors, cancer-related variables (including treatments, body image concerns, and systemic chronic inflammation), psychological and behavioral factors, and immunological responses interact bidirectionally, creating a complex network that predisposes to and maintains insomnia. Hormonal alterations, particularly activation of the hypothalamic–pituitary–adrenal axis by stressors and the sex-specific impact of inflammatory and immune mediators, are central mechanisms, with menopause acting as a key modulator. Menopause contributes to insomnia through several downstream consequences: neuronal modulation (involving KNDy neurons and neurotransmitters such as serotonin, dopamine, and GABA), vasomotor symptoms with associated changes in mood, cognition, and “brain fog”, and other menopausal issues (including dysmetabolism, weight gain, genito‑urinary syndrome, sexual dysfunction, fatigue, and additional symptoms). Altogether, these interconnected biological, psychological, and treatment‑related factors reinforce each other to perpetuate insomnia in women with cancer.
Comment 6:
- Please, ensure that all abbreviations used in the text appear in the abbreviations list.
Thank you for your suggestion. We revised the abbreviations list.
Comment 7:
- The section on treatment is informative and balanced. However, the review would benefit from a more explicit clinical perspective. When should oncologists screen or refer patients to sleep specialists? How should sleep disturbances be integrated into survivorship care pathways?
According with your suggestion, we added the following paragraph at the end of Treatment section.
Drawing on the evidence summarized above and in line with current guideline recommendations, oncologists should first be familiar with the most prevalent sleep disorders in patients with cancer, particularly insomnia. This knowledge is essential because, in routine clinical practice, oncologists should systematically assess the presence of sleep disturbances in their patients. When adequately trained and confident in managing these conditions, oncologists may initiate first‑line interventions for uncomplicated insomnia, while referring patients with complex or treatment‑refractory insomnia, as well as those with suspected alternative sleep disorders, to a sleep specialist. Ideally, a sleep expert should be embedded within the multidisciplinary oncology team; when this is not feasible, there should at least be a designated sleep center to which patients can be referred for comprehensive assessment and management.
Comment 8:
- The authors discuss melatonin, DORAs, MHT, phytoestrogens, and NK receptor antagonists. However, It would be helpful to clearly distinguish which therapies have evidence specifically in cancer survivors and where data are extrapolated from non-oncologic populations.
We sincerely thank you for your valuable suggestions and greatly appreciate your insightful proposals. For melatonin, we have already stated that ‘randomized controlled trials are limited by small sample sizes, high risk of bias, and indirectness, with only a few studies specifically addressing breast cancer populations,’ whereas for DORAs we noted that ‘to date, no clinical studies have been conducted on dual orexin receptor antagonists (DORAs) specifically in patients with female cancers.’
Regarding gynecological treatments such as MHT and phytoestrogens, the references pertain to cancer survivors, with specific reference to gynecologic neoplasms, and, despite limited published evidence, to recently published studies on the use of hormones in this population. Concerning NK receptor antagonists, we have clarified that in women with hormone-responsive cancers their use remains off-label, and we have also updated information regarding the recent approval of Elinzanetant for breast cancer survivors.
Reviewer 2 Report
Comments and Suggestions for Authors
The authors have reviewed how insomnia in women surviving breast and gynecological cancers is a multidimensional issue influenced by hormonal factors. The hormonal changes are triggered by cancer treatment protocols. The review emphasizes the need for comprehensive treatment protocols that effectively manage insomnia and its related conditions in women surviving breast and gynecological cancers.
Insomnia is often associated with many other types of sleep disorders. The authors did not discuss whether insomnia and its extent differ between the breast cancer and gynecological cancer surviving women.
Would it be appropriate to replace ‘insomnia’ with ‘sleep disorders’ in the title of the review article? Suggested title: “Hormonal Influences on Sleep Disorders in Women Surviving Breast and Gynecologic Cancers: A Narrative Review”.
However, one important aspect that could have been included is the bibliometric approach the authors have used to identify and select the relevant literature for the review.
Author Response
Comment 1:
The authors have reviewed how insomnia in women surviving breast and gynecological cancers is a multidimensional issue influenced by hormonal factors. The hormonal changes are triggered by cancer treatment protocols. The review emphasizes the need for comprehensive treatment protocols that effectively manage insomnia and its related conditions in women surviving breast and gynecological cancers.
Insomnia is often associated with many other types of sleep disorders. The authors did not discuss whether insomnia and its extent differ between the breast cancer and gynecological cancer surviving women.
We truly appreciate your contribution and suggestion. Unfortunately, based on current knowledge, it is not possible to clearly define whether there are real differences between women with breast cancer and those with gynecologic malignancies, as the available literature is limited. We will certainly emphasize the need for further studies to better stratify prevalence, impact, and treatment across different cancer types, within the framework of personalized medicine.
Comment 2:
Would it be appropriate to replace ‘insomnia’ with ‘sleep disorders’ in the title of the review article? Suggested title: “Hormonal Influences on Sleep Disorders in Women Surviving Breast and Gynecologic Cancers: A Narrative Review”.
We sincerely appreciate your thoughtful suggestion regarding the title. We chose to focus specifically on insomnia, as it represents the most common sleep disorder, and we deliberately limited the review to this condition in order to present the most homogeneous data possible, given their limited availability. We hope it will be agreeable to maintain the current title, as it emphasizes insomnia in oncological patients while providing a specific perspective on gynecologic concerns, especially regarding hormonal impact. We believe this focus allows for a clear and meaningful contribution to the literature, while still acknowledging the broader context of sleep disturbances in this population.
Comment 3:
However, one important aspect that could have been included is the bibliometric approach the authors have used to identify and select the relevant literature for the review.
As also suggested by the other reviewer, we added a paragraph to improve methodological transparency and strengthen the rigor of the narrative review.
- Search Strategy
Although this is a narrative review and not a systematic one, we adopted a structured search strategy to ensure transparency. Relevant literature was identified through PubMed/MEDLINE, Scopus, and Web of Science using combinations of terms such as insomnia, sleep disorders, sleep quality, breast cancer, gynecologic cancer, menopause, and iatrogenic menopause. The search covered primarily 2000–2024, with inclusion of earlier landmark studies when necessary. We prioritized peer-reviewed clinical studies, systematic reviews, meta-analyses, and major guidelines. Evidence was qualitatively synthesized across oncology, gynecology, endocrinology, and sleep medicine, with a focus on mechanisms, clinical impact, and management of insomnia in female cancer survivors.